

# 1    Impact of wildfires on Canada's oilsands facilities

**Nima Khakzad**
Faculty of Technology, Policy, and Management, Delft University of Technology, Delft 2628BX, The
Netherlands
Correspondence to: Nima Khakzad  (n.khakzadrostami@tudelft.nl)
**Abstract.**
Exponential growth of oil and gas facilities in wildlands from one side and an anticipated increase of global
warming from the other have exposed such facilities to an ever-increasing risk of wildfires. Extensive
oilsands operations in Canadian wildlands especially in the Province of Alberta along with the recent
massive wildfires in the province requires the development of quantitative risk assessment (QRA)
methodologies which are presently lacking in the context of wildfire-related technological accidents. The
present study is an attempt to integrate Canadian online wildfire information systems with current QRA
techniques in a dynamic risk assessment framework for wildfire-prone process plants. The developed
framework can easily be customized to other process plants potentially exposed to wildfires worldwide
provided that the required wildfire information is available.
**Keywords:** Wildfires; Process plants; Domino effect; Quantitative risk assessment; Natech accidents



**Nomenclature**
API: American Petroleum Institute
BUI: buildup index
D: flame depth
DC: drought code
DMC: duff moisture code
FBP: fire behavior prediction
FFMC: fine fuel moisture code
FWI: fire weather index
$F_{view}$: view factor
h: flame height
H: fuel's low heat of combustion
HFI: head fire intensity
ISI: initial spread index
L: flame length
P(.): marginal damage probability of target vessel
P(.|w): conditional damage probability of target vessel given a wildfire
$P_{arr}$: probability of a smoldering fire escalating to a flaming fire
$P_B$: burn probability
$P_{ign}$: probability of ignition given a long-continuing current
$P_I$: probability of ignition
$P_{LCC}$: probability of a long-continuing current
$P_{sur}$: probability that a smoldering ignition survives
$P_w$: probability of wildfire
Q: reaction intensity
$Q_x$: heat radiation at the distance of x
r: fire's rate of spread in the direction of the fire head
ROS: rate of spread
ttf: time to failure of target vessel
V: volume of target vessel
w: fuel's combustion rate in the flaming zone
WIPP: wildfire ignition probability predictor
x: horizontal distance from the flame's centre



Y: probit value
σ: probability of a tree's self ignition
θ: probability of fire spread from one tree to the others
λ: probability of tree growth in an empty cell
$\tau_a$: atmospheric transmissivity
ϕ: cumulative standard normal distribution



## 1. Introduction


Weather-related disasters, especially heatwaves, wildfires, droughts, floods and hurricanes have been
foreseen to affect around two-thirds of the European population annually by the end of this century
(Forzieri et al., 2017). Canada and the U.S. are no exception as evident by the recent hurricanes, floods, and
wildfires which devastated the states of Texas and California in the U.S. and the provinces of British
Columbia and Alberta in Canada. Aside from the impact of such natural disasters on the environment and
urban areas, their effect on industrial plants and hazardous facilities (process plants, nuclear plants, etc.)
has started to raise concerns in academia, the industry, and regulatory bodies.
Massive fires in a refinery in Turkey in 1999 during the Kocaeli earthquake, substantial release of
petroleum products and chemicals in the U.S. during Hurricane Katrina in 2005 and Hurricane Harvey in
2017, extensive damage to coastal industrial complexes in Japan in 2011 during the Great Sendai
Earthquake and the following tsunami, and shut-down of oilsands plants which incurred enormous oil
production losses during massive wildfires in Canada in 2016 are just some examples among the others.
Although the hazard of wildfires in ecological and urban risk assessment studies has long been recognized
(Preisler et al., 2004; Beverly and Bothwell, 2011; Scott et al., 2012, 2013; Lozano et al., 2016), the relevant
work in the context of wildland-prone industrial complexes has been very limited, if any. In Europe, for
example, Seveso Directive III (2012) has only recently mandated the member states to consider the
probability of natural disasters in the risk assessment of major accident scenarios when preparing safety
reports (Article 10), with an explicit mention of floods and earthquakes (the Annex II) but the wildfires.
The most of European countries that consider natechs have likewise limited their focus to only a few
natural hazards (Krausmann and Baranzini, 2012). Table 1 exemplifies some of such efforts.
Exponential growth of industrial facilities and the subsequent prolongation of wildland-industry interfaces
from one side and an anticipated increase of global warming from the other are expected to increase the
frequency and severity of technological accidents caused by natural disasters, including the wildfires.
In May 2015, a massive wildfire in northern Alberta, Canada, spread into the oilsands areas, threatening
several operations and keeping about 10% of the production offline. Two major petroleum companies,
Canadian Natural and Cenovus Energy, shut down their 80,000 and 135,000-barrel-a-day operations,
respectively, for safety precautions as the fires approached Foster Creek oilsands facility and Caribou South
natural gas plant (Mining.Com, 2015).



**Table 1**. Natural hazards considered in safety assessment and management of process plants in European Union
(Krausmann and Baranzini, 2012).

| Country | Natural hazard |
| --- | --- |
| Lithuania | Floods |
| Slovakia | Floods |
| Czech Republic | Mainly floods |
| UK | Mainly floods |
| Romania | Floods, landslides, earthquakes |
| Germany | Floods, storms, earthquakes |
| France | Floods, landslides, earthquakes, lightning |
| Italy | Floods, storms, earthquakes, lightning, wildfire |
| Netherlands | All-hazards approach* |

* It is not identified whether it accounts for wildfires.
In May 2016, a wildfire burned part of Fort McMurray, Alberta, Canada, and spread towards oilsands plants
north of the city where major oilsands production plants Syncrude and Suncor Energy along with some
smaller petroleum operations were located, resulting in a 40% drop in production at nearby oilsands
facilities (Figure 1).

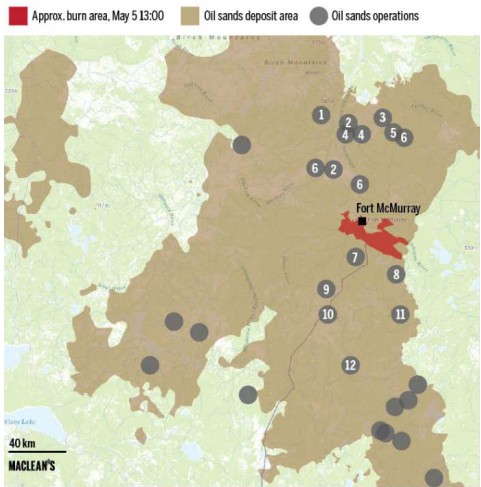


**Figure 1**. Wildfire in Fort McMurray and the location of affected oilsands plants: ① Canadian Natural Resources, ②
Syncrude joint venture, ③ Imperial Oil, ④ Shell Canada, ⑤ Husky Energy/BP, ⑥ Suncor, ⑦ Athabasca, ⑧ Nexen
(CNOOC), ⑨ Japan Canada Oil Sands, ⑩ Connacher Oil and Gas, ⑪ ConocoPhillips, ⑫ Statoil (Maclean's, 2016a).



The operations shutdowns or reductions were also influenced by precautionary shutdowns of pipeline
carrying diluent, a flammable substance needed to thin the oilsands bitumen, resulting in roughly as much
as one million barrels a day reduction of the oilsands' output (Maclean's, 2016a). The wildfire did not cause
damage to oilsands plants and process equipment, but it burned down a 665-unit worker accommodation
camp northern Fort McMurray (Global News, 2016a). But what would have happened if the fire had
reached the oilsands mines and the production facilities?
As far as it concerns the oilsands mines, bitumen, the main component of oilsands, does not easily catch fire
(Global News, 2016b). Considering the fact that 80% of bitumen is buried deep underground, the bitumen
in oilsands mines is mixed with sand (similar to asphalt), and would probably smolder if ignited (Maclean's,
2016b). However, oilsands projects rely on two highly flammable substances for the extraction, process,
and transport the bitumen: Natural gas and diluent, which is a very light petroleum substance.
Natural gas is used to generate power for the plants and heat up the steam used to liquefy the bitumen.
Diluent, on the other hand, is used to dilute the crude bitumen thin enough to flow through pipelines. Both
the natural gas and diluent can pose high risks if exposed to fire though the pipes carrying them are usually
buried underground.
Oilsands process plants are usually accompanied by large tank terminals in the vicinity to store oil
products. Exposed to external fires (such as wildfire), buckling of atmospheric storage tanks and spill of
hydrocarbons, tank fires, vapor cloud explosions, and explosion of pressurized tanks can be recognized as
potential risks (Heymes et al., 2013, Godoy 2016). In case one or more storage tanks are ignited by the
wildfire, the tank fire(s) can impact adjacent storage tanks, leading to a fire domino effect.
In order to protect oilsands facilities from wildfires (and also protect the forest from potential ignition
sources at the facilities), there is a buffer zone (safety distance in the form of vegetation-free ground)
between facilities and forest vegetation. In the absence of methodologies to quantify the risk imposed by
wildfires, such buffer zones are usually determined based on rule-of-thumb guidelines (e.g., see FireSmart,
2012). Numerical simulations of storage tanks exposed to wildfire has, however, demonstrated that in the
most cases such safety distances would not suffice (Heymes et al., 2013).
Due to extensive oilsands operations in Canadian wildlands, in the present study, we have developed a
dynamic framework, mainly based on available techniques and daily updated wildfire maps made available
online by Government of Canada, to assess the impact of wildfires on oilsands facilities. Since the
framework is modular, it can be tailored to assess the risk of wildfires at process plants in wildfire-prone
areas worldwide. Section 2 revisits the Canadian wildland fire information system; in Section 3, the



components of wildfire risk assessment are described and quantified; Section 4 is devoted to the impact
assessment of wildfires on process facilities; Section 5 concludes the study.

## 2. Canadian Wildfire Information System

In Canada, two systems are being used to determine the characteristics and the hazard of wildfires:
Canadian Forest Fire Weather Index System, and Canadian Forest Fire Behavior Prediction System. The
former is mostly concerned with the estimation of wildfires' basic components (e.g., flammability of
vegetation) whereas the latter deals with the dynamics of wildfires (e.g., fire intensity). Since in the present
study the identification and quantification of wildfires in Canadian wildlands are mainly based on the
foregoing two systems, they will be recapitulated in this section.

### 2.1. Forest Fire Weather Index System

Wildfires, like other types of fire, can be defined using the fire triangle consisting of fuel (trees, grasses,
shrubs), oxygen, and heat source. As much as it concerns the fuel, parameters such as the fine fuel moisture
code (FFMC), which is the moisture content of litter and other crude fire fuels, duff moisture code (DMC),
which is the moisture content of loosely compacted organic layers of moderate depth and woody materials,
and drought code (DC), which is the average moisture content of deep compact organic layers and large
logs, are taken into account to determine both the ease of ignition and the flammability of the available fuel.
DMC and DC are combined together to determine the total amount of combustible materials in the form of a
so-called buildup index (BUI). Accordingly, the wind and the FFMC are combined to predict the rate of fire
spread in the form of a so-called initial spread index (ISI). Having the BUI and the ISI, the fire weather index
(FWI), as an indication of fire danger, can be determined as shown in Figure 2 (Natural Resources Canada).
Figure 3(a) illustrates the fire weather index (FWI) of Canada on May 1, 2016, a day before the Fort
McMurray wildfire. Based on the FWI and the type of fire (surface fire, crown fire, intermittent crown
involvement), the fire danger index can be determined (low, moderate, high, very high, extreme) as an
indication of how easy it is to ignite the forest fuel, how difficult it is to control the fire, and the type of
firefighting equipment needed (pumps, tanker trucks, bulldozer, aircraft, etc.) as shown in Figure 3(b).



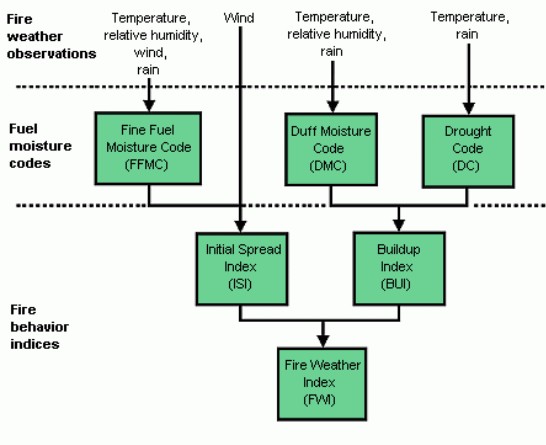


**Figure 2**. Identification of fire weather index (Natural Resources Canada)

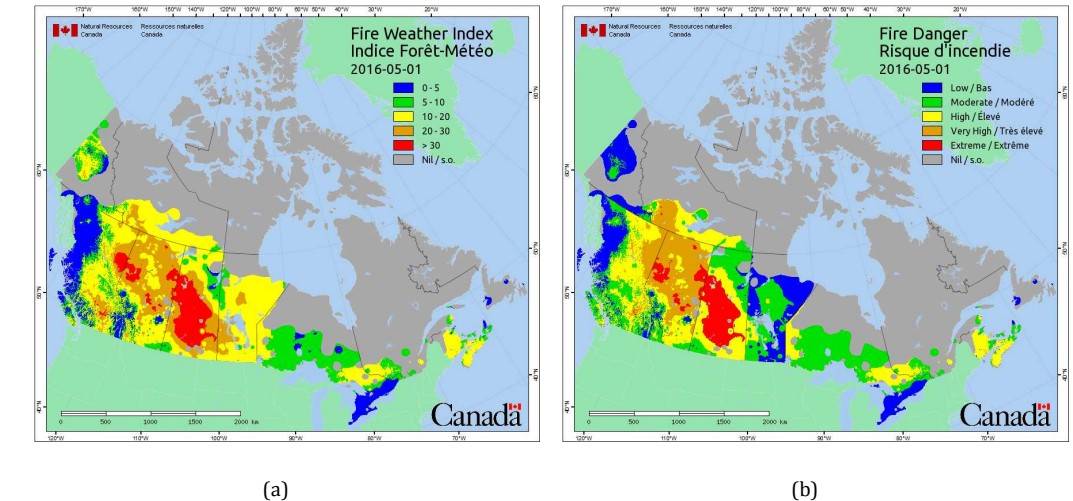


(a) (b)
**Figure 3**. (a) Fire weather index, and (b) Fire danger index of Canada on May 1, 2016 (Natural Resources Canada).
**2.2. Forest Fire Behavior Prediction System**
To quantify the impact of wildfires on industrial plants, quantitative estimates of head fire spread rate, fuel
consumption, and fire intensity are needed. Canadian Forest Fire Behavior Prediction System employs an
elliptical fire growth model (Tymstra et al., 2010) to estimate the fire area, perimeter, perimeter growth




rate, and flank and back fire behavior. The rate of spread (ROS) is the predicted speed (m/min) of the fire
head (fire front), which is calculated based on the fuel type, initial spread index (ISI), buildup index (BUI),
crown base height, and other parameters (Natural Resources Canada).
Head fire intensity (HFI) is an estimate of the energy output per meter of the fire front (kW/m), calculated
based on the rate of spread (ROS) and total fuel consumption ($kg/m^2$). The rate of spread (ROS) and head
fire intensity (HFI) indices calculated by the Canadian Wildland Fire Information System a day before the
start of the Fort McMurray wildfire are shown in Figures 4(a) and (b), respectively (Natural Resources
Canada).
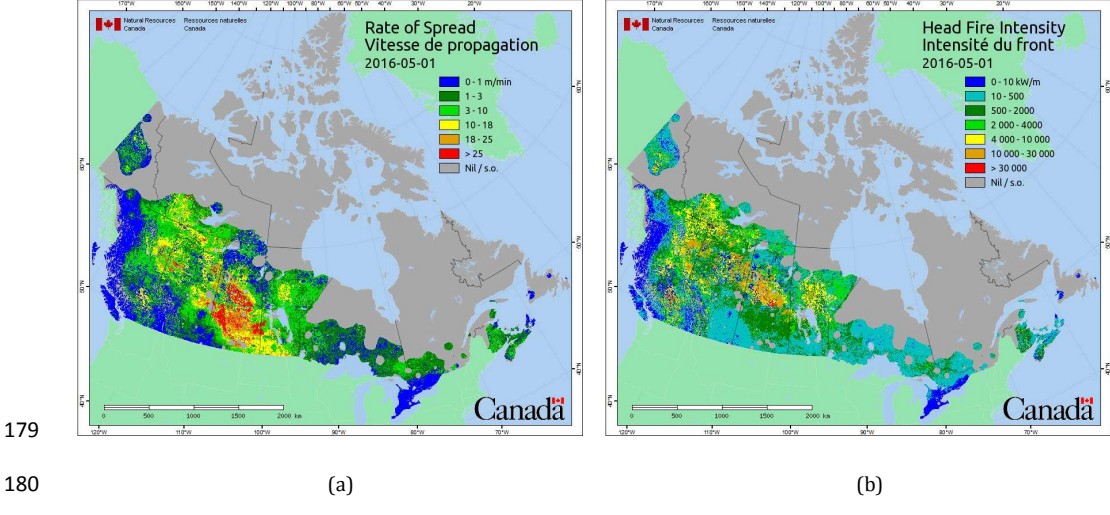

180                            (a)                                                    (b)

**Figure 4**. (a) Fire rate of spread, and (b) Head fire intensity in Canada on May 1, 2016 (Natural Resources Canada).
## 3. Wildfire Risk Assessment
In wildfire risk assessment, the ignition probability, burn probability (the probability that wildfire reaches
to a certain spot), type of fire (surface fire, crown fire, intermittent crown involvement) and fire intensity
are the main factors to take into account (Scott et al., 2013).
Many methodologies have been developed to predict the ignition probability (Latham and Schlieter, 1989;
Lawson et al., 1994; Anderson, 2002), to model surface fire spread (Rothermel, 1972), crown fire spread
(Rothermel, 1991), and transition between surface and crown fire spread (van Wagner, 1977). Accordingly,
a number of software tools such as FARSITE (Finney, 1998), FlamMap5 (Finney, 2006), FSPro (Finney et al.,



2011a), and FSim (Finney et al., 2011b) have been developed based on historical records of regional
wildfires, weather conditions, type and density of vegetation in the landscape, and the topology of the
landscape. Using the developed models and software tools, the risk imposed by wildfires on an oilsands
facility can be modeled as the product of the wildfire probability, $P_W$, and the severity of consequences,
preferably in monetary units as:
Wildfire's risk = $P_W$ . Consequence (1)
Given the geographical location of the facility, the probability of wildfire at the borders of the facility can be
estimated as the probability of having a small fire somewhere at the landscape ($P_I$) times the probability of
the small fire growing to a wildfire larger than 400 m² in area and reaching the location of the facility ($P_B$):
$P_W = P_I$ . $P_B$ (2)
$P_I$ and $P_B$ are also known as ignition probability and burn probability, respectively. Exposed to a wildfire,
the potential consequences and their severity depend on the wildfire intensity and the facility's
vulnerability to wildfire: C = f (fire intensity, facility's vulnerability)[1]. In the following sections we will
describe the components of wildfire risk in further detail and explain how they can be estimated or
acquired from available (mostly freely accessible) models and databases, with a particular emphasis on
Canadian forest fire system.

### 3.1. Ignition probability

Wildfires can be categorized as hydro-geological events which are bound to increase especially due to
global warming. Every degree in warming increases the possibility of lightning, which is one of the major
triggers of wildfires, by 12% (Romps et al., 2014). Likewise, 15% more precipitation would be needed to
offset the increased risk of wildfires due to one degree increment of warming (Flannigan et al., 2016).
Nevertheless, man-made fires (burning campfires, cigarettes) account for 80% of wildfires (National
Geographic).
Weather conditions such as temperature, relative humidity, and wind speed are key factors in the
probability estimation of an ignition (small fire) which can lead to a wildfire. In addition to the weather
conditions, the vegetation moisture content (equal to FFMC) plays a key role not only in the initiation of fire
(the ignition probability) but also in the continuation and spread of fire (fuel flammability) (Chuvieco et al.,

217 2004).

---

[1] In the present study, we do not consider the indirect risk incurred by, among others, loss of production due to plant's precautionary shutdowns, staff evacuation, or the like.





Based on the measurement of FFMC in consecuitive time periods before the start of a potential wildfire, the
logistic regression has been used to roughly predict $P_I$ based on FFMC (Larjavaara et al., 2004; Jurdao et al.,
2012). Similarly, Preisler et al. (2004) used the logistic regression to predict the probability of small fires
(equivalent to $P_I$) based on parameters such as burning index, fire potential index, drought code, wind
speed, relative humidity, dry bulb temperature, day of the year, and the elevation.
Lawson et al. (1996) developed an application called Wildfire Ignition Probability Predictor (WIPP) to
predict, on an hourly or daily basis, the $P_I$ of man-made wildfires in British Columbia forests, Canada. Based
on the calculations of FFMC and 10-meter wind speed, WIPP estimates $P_I$ in three categories as low (0-
50%), medium (50-75%), and high (75-100%). Considering the lightning as one of the main triggers of
wildfires, Canadian Wildland Fire System estimates the time-dependent probability of lightning-caused
ignitions as (Anderson 2002):
$$P_I = P_{LCC} \cdot P_{ign} \cdot P_{sur} \cdot P_{arr} \qquad (3)$$
where $P_{LCC}$ is the probability of a long-continuing current (85% for positive flashes, 20% for negative
flashes across Canada); $P_{ign}$ is the probability of ignition given a long-continuing current, determined by
fuel type, forest floor depth, and moisture conditions (Latham and Schlieter 1989; Anderson 2002); $P_{sur}$ is
the probability that a smoldering ignition will continue to survive as a smoldering fire, determined by the
fuel moisture, the bulk density, and the inorganic content of the forest floor (Hartford 1989; Anderson
2002); $P_{arr}$ is the probability of a smoldering fire escalating to a flaming fire (Lawson et al. 1994; Forestry
Canada Fire Danger Working Group 1992; Anderson 2002).
Wildfire-prone provinces in Canada such as Alberta and British Columbia provide ignition probability maps
on a daily basis both for the current day and the next day. Figure 5 depicts the $P_I$ map for the Province of
Alberta administrated by Alberta Agriculture and Forestry.




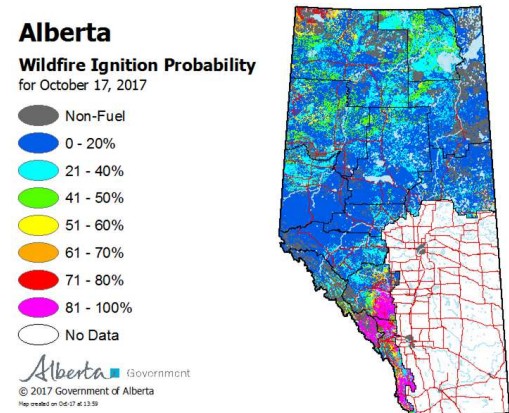


242          **Figure 5**. Wildfire ignition probability ($P_I$) in Alberta, Canada (http://wildfire.alberta.ca)

**3.2. Burn probability**
Burn probability ($P_B$) is the conditional probability that a small fire somewhere in the landscape would
escalate to a wildfire and burn somewhere else in the landscape. Estimation of $P_B$ is challenging as the
spread of wildfire from one point to another is a complicated process affected by many factors such as the
type of vegetation (fuel), weather conditions, and land topology. These factors, in turn, consist of several
key parameters such as the flammability of fuel, vertical arrangement of fuel, moisture content of fuel, wind
speed and direction, relative humidity, the oriantion of fire (downhill or uphill), the type of fire (surface
fire, crown fire, surface-crown transition), etc.
Considering the foregoing fire spread parameters, $P_B$ can be estimated as the relative frequency of
wildfires' burning a certain spot given a number of small fires at different spots of the landscape (Scott et
al., 2013). Models developed for wildfire spread simulation include empirical, semi-empirical, and physical
models (Pastor et al., 2003). Some of these models such as FARSITE[2] (Finney, 1998) and BehavePlus
(Andrews, 2013) need detailed spatial information on topography, fuels, and weather conditions, not
readily available for many locations of interest. A comprehensive review of wildfire simulation models can
be found in Papadopoulos and Pavlidou (2010). Less sophisticated models and software have also been
developed for fire spread modeling and investigating whether a small fire at point A would evolve as a
wildfire at point B in the landscape.
Drossel and Schwabl (1992) developed a simple forest-fire model based on the following assumptions:

---

[2] FARSITE is available from https://www.firelab.org/project/farsite.



• considering the landscape as a grid, each cell (A) can have three states: "empty", "occupied by tree",
and "burning tree", that is, A = (empty, tree, burning).
• fire from a burning cell can spread with a probability of θ to other cells in its Moore neighborhood
(i.e., at most eight other cells). In other words, if cell B is in the Moore neighborhood of cell A, $P(B^{t+1}$
$= burning \mid B^t = tree, A^t = burning) = θ$.
• a cell can ignite with a probability of σ (self ignition probability) even if no other cells in its
neighborhood are on fire; that is, $P(B^{t+1} = burning \mid B^t = tree, A^t = tree) = σ$.
• an empty cell can be filled with a probability of λ with a tree (usually considered if time between
two sequential fires would be long enough to allow for growing new plantation). In other words,
$P(B^{t+1} = tree \mid B^t = empty) = λ$.
Fire spread models can be coupled with Monte Carlo simulation to estimate $P_B$. For instance, Figure 6
depicts the output of the forest-fire model encoded in a Javascript program[3], where a random small fire
ignited somewhere south of the landscape (Figure 6(a)) evolves to a wildfire (Figure 6(b)). Assuming that
the process facility of interest (e.g., oilsands plant or oil terminal) is located in the north of the landscape,
the probability of the wildfire reaching the facility (cells) north of the landscape can thus roughly be
estimated as:
$$P_B = \frac{n}{N} \tag{4}$$
where N is the total number of Monte Carlo simulations, that is, the total number of random small fires at
different spots of the landscape; n is the total number of simulations where a small fire turned out as a
wildfire and reached the north of the landscape (Figure 6(c)).

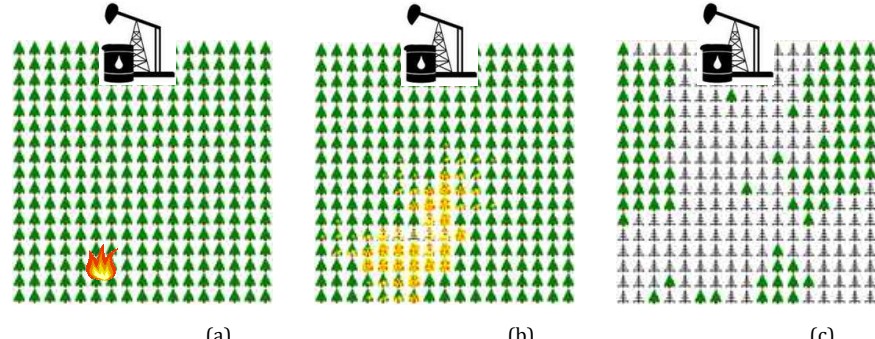

(a)          (b)          (c)
**Figure 6**. Wildfire spread in a hypothetical landscape. (a) Ignition of small fire south of the landscape. (b) The small
fire escalates as a wildfire. (c) The wildfire reaches the process facility north of the landscape.

---

[3] The program is available from http://www.shodor.org/interactivate/activities/Fire/.



Similar attempts have been made, for example, using NetLogo (Wilensky, 1997), which is a multi-agent
programmable modeling environment, to model fire spread yet based on simplistic assumptions and a
limited number of parameters (e.g., density of trees).

**3.3. Fire intensity**

Head fire intensity (HFI) is the rate of heat release per unit length of the fire head (kW/m), regardless of
the fire's depth. HFI, which is also known as Byram's fire intensity or frontal fire intensity, can be calculated
as (Byram, 1959):
HFI = H. w. r (5)
where H (kJ/kg) is the fuel's low heat of combustion, w (kg/m²) is the fuel's combustion rate in the flaming
zone, and r (m/s) is the fire's spread rate in the direction of the fire head (Figure 7). H is equal to the high
heat of combustion minus the heat losses from radiation, incomplete combustion, and fuel moisture.
Compared to the other parameters in Byram's fire intensity, H varies slightly from fuel to fuel and can thus
be considered as a constant. Alexander (1982) suggests a basic value of 18700 kJ/kg.

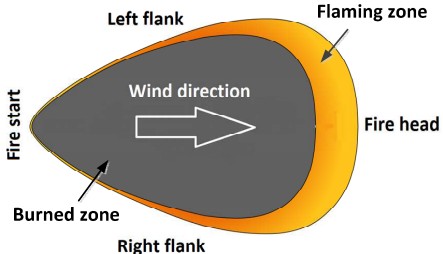


**Figure 7**. Different zones of a wildfire (adapted from Wikipedia).


Values of r and w, however, can vary significantly for different fuels. Considering r, for instance, a grass fire
may travel at a rate of r = 5 km/h whereas fire in a dry eucalypti forest may travel at a rate of r = 1 km/h
capable of throwing embers up to 1 km ahead of the fire (Cheney, 1990). As a result, HFI can vary from 15
to 100,000 kW/m (Byram, 1959) though it rarely exceeds 50,000 kW/m, and for the most of crown fires
lies in the range of 10,000–30,000 kW/m (Alexander, 1982). Having the flame length, L(m), Byram (1959)
has suggested Equation (6) to calculate the HFI of surface fires:
$HFI = 260\ L^{2.174}$ (6)



In case of crown fires, one-half of the mean canopy height should be added to L (Byram, 1959). Flame
length (L), flame height (h), and the flame depth (D) have been depicted in Figure 8. At very low wind
speeds on level terrain, h and L can be considered the same. A thorough review of developed relationships
to calculate the fire intensity based on the fire length can be found in Alexander and Cruz (2012).
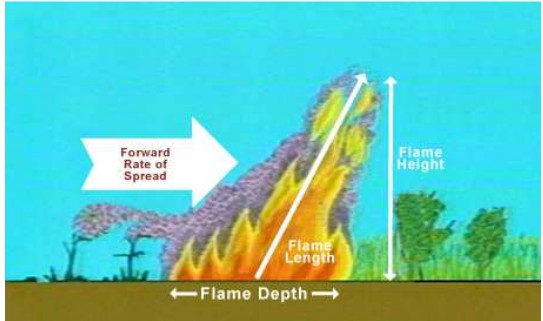
**Figure 8**. Flame characteristics (Utah State University website).
Based on the flame length (L), the fire intensity (HFI) can also be classified into six classes (Scott et al.,
2013) as listed in Table 2; this way, the observations of L can be used to make rough estimates of HFI.
**Table 2**. Flame length range associated with six standard fire intensity classes.

| Fire intensity class | Flame length (m) |
|---|---|
| Class 1 | 0.0 - 0.6 |
| Class 2 | 0.6 - 1.2 |
| Class 3 | 1.2 - 1.8 |
| Class 4 | 1.8 - 2.4 |
| Class 5 | 2.4 - 3.7 |
| Class 6a | 3.7 - 15 |
| Class 6b | > 15 |


The fire intensity classes in Table 2 can be associated with the wildfire ranks used by the British Columbia
Wildfire Service[4] for a quick description of fire behavior based on wildfire visual observations (Table 3).
Similar classes as of Tables 2 and 3 are also provided by Canadian wildfire protection agencies such as
Alberta Wildfire (Figure 9), which accordingly can be used to infer the flame length (L) using Table 2 and
then to estimate the fire intensity (HFI) using Equation (6). As another option, the head fire intensity maps
provided by the Canadian Wildfire System (Figure 4(b)) can be used to directly identify the HFI.

---

[4] https://www2.gov.bc.ca/gov/content/safety/wildfire-status/about-bcws/wildfire-response/fire-characteristics/rank



**Table 3**. Wildfire ranks used by the British Columbia Wildfire Service to determine the fire intensity.

| Visualization | Rank | Description | Characteristics |
|---|---|---|---|
| 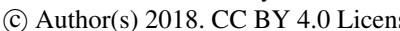 | 1 | Smouldering ground fire | • Smouldering ground fire<br>• No open flame<br>• White smoke<br>• Slow (i.e. creeping) rate of fire spread |
|  | 2 | Low vigor surface fire | • Surface fire<br>• Visible, open flame<br>• Unorganized or inconsistent flame front<br>• Slow rate of spread |
|  | 3 | Moderately vigorous surface fire | • Organized flame front – fire progressing in organized manner<br>• Occasional candling may be observed along the perimeter and/or within the fire<br>• Moderate rate of spread |
|  | 4 | Highly vigorous surface fire with torching, or passive crown fire | • Grey to black smoke<br>• Organized surface flame front<br>• Moderate to fast rate of spread on the ground<br>• Short aerial bursts through the forest canopy<br>• Short-range spotting |
|  | 5 | Extremely vigorous surface fire or active crown fire | • Black to copper smoke<br>• Organized crown fire front<br>• Moderate to long-range spotting and independent spot fire growth |
|  | 6 | A blow up or conflagration; extreme and aggressive fire behaviour | • Organized crown fire front<br>• Long-range spotting and independent spot fire growth<br>• Possible fireballs and whirls<br>• Violent fire behaviour probable<br>• A dominant smoke column may develop which influences fire behaviour |

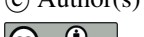



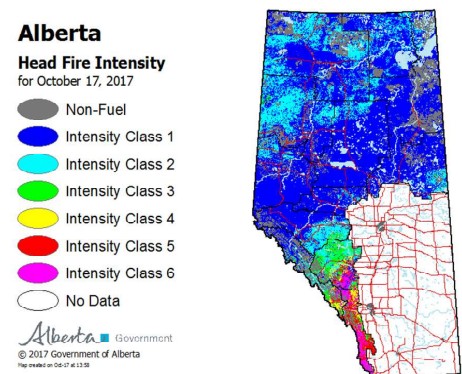


**Figure 9**. Wildfire intensity classes in Alberta, Canada (http://wildfire.alberta.ca)


Having the flame depth (D), the frontal fire intensity (HFI) can be converted to area-fire or reaction
intensity Q (kW/m²) (Alexander, 1982):
$Q = \frac{HFI}{D}$ (7)
Considering the flame as a solid body (Butler and Cohen, 2000; Heymes et al., 2013), the amount of reaction
intensity at a distance of *x* from the flame's ground centre (see Appendix) can be calculated using Solid
Flame Model (Mudan, 1987) as:
$Q_x = Q \cdot F_{view} \cdot \tau_a$ (8)
where $F_{view}$, the view factor, is the fraction of the heat radiation received by a receptor (Assael and
Kakosimos, 2010), and $\tau_a \in [0, 1]$ is the atmospheric transmissivity, corresponding to the fraction of the
thermal radiation received by the receptor considering the mitigation effect of humidity and carbon dioxide
as well as the dissipation due to the distance. In the determination of safety zones, $\tau_a = 1$ is used for
conservative results (Heymes et al., 2013).
**4. Impact of wildfire on oil storage tanks**
During wildfires, the main threats to oilsands facilities – either the process plant or the storage terminal –
come from airborne embers and radiant heat. The threat of airborne embers is even greater since they are
able to travel with wind for several kilometers ahead of the fire front. The accumulation of airborne embers
near tank openings and vents or under the base of structures and process vessels, given enough vegetation
or spilled flammable hydrocarbons, can ignite a fire – also known as spotting  (FireSmart, 2012) – which



may easily escalate to a major fire and possibly a domino effect given the large inventory of flammable
substances stored in the facility.
Assessing the risk of wildfire's embers is very tricky considering several influential parameters such as the
direction and speed of the wind, the trajectory of embers, the accumulation of embers near critical spots,
availability of onsite vegetation or spilled hydrocarbons, whose prediction is subject to large uncertainties
if not impossible. Despite the difficulties in impact assessment of wildfire embers, simple protection and
mitigation measures can be taken to effectively reduce their threat. For instance, limiting the use of floating
roof tanks as the most common type of tanks reportedly involved in tank fires (Godoy, 2016), encouraging
the use of cone roof tanks to prevent embers from landing around openings and vents, turning the vents
downward and covering the openings with wire mesh, removing vegetation around tanks and combustible
structures, and equipping the structures and storage tanks with sprinkler systems, are some of the
measures to tackle the risk of airborne embers (FireSmart, 2012).
Aside from the impact of embers, the radiant heat emitted from the wildfire can threat the integrity and
safety of process vessels and storage tanks. The type and severity of such impact depend on the intensity of
the radiant heat received by target vessels as well as their type (atmospheric, pressurized, pipeline, etc.)
and dimension (usually their volume). Radiant heat acts as a thermal load on the wall of the vessels, which
are categorized as thin-walled structures, and affects the stiffness and strength properties of the wall
material (usually steel, in the oil and gas industry).
In the case of atmospheric storage tanks such as oil and gasoline tanks, this change in properties results in
wall weakening and is usually followed by large radial displacements in the form of buckling (Godoy,
2016). Buckling of steel storage tanks subject to thermal loading has thoroughly been investigated in Liu
(2011) and Mansour (2012). A review of oil storage steel tanks under different types of loads, including
thermal loading, can also be found in Godoy (2016). Exposed to external fires, empty or partially filled
storage tanks may receive up to five times higher temperature than completely filled tanks, and thus more
susceptible to buckling. For partially filled tanks, there is even a jump between the temperature below and
above the liquid level (Liu, 2011).
In addition to the possibility of buckling, which endangers the integrity of storage tanks, petroleum
products may ignite spontaneously at their autoignition temperatures in normal atmosphere without even
direct impingement of wildfire flames or airborne embers. Autoignition temperature of most of petroleum
products is between 200 to 250 degrees Celsius, well below the temperature required for buckling of steel
storage tanks and easily reachable for storage tanks exposed to radiant heat of wildfires. For intact



atmospheric storage tanks, the autoignition of flammable contents would most probably lead to tank fires
while for damaged storage tanks with spilled fuel in the catch basins it would lead to pool fires.
For pressurized tanks such as LPG[5] tanks, on the other hand, BLEVE[6] is the most likely scenario. BLEVE
occurs when the increase in the internal vapor pressure of the tank exposed to an external fire grows
beyond the strength of the already weakened tank wall, leading to the formation of a tear. If the tear
spreads to the entire length of the tank a BLEVE occurs, followed by a fireball; otherwise, a jet fire would be
expected (Birk and Cunningham, 1994). In order to prevent from the increase in the internal overpressure,
pressurized tanks are usually equipped with pressure relief valves or fusible plugs, which are nevertheless
likely to damage and fail to operate (CSB, 2008). Furthermore, to prevent from BLEVE, the American
Petroleum Institute (API) has identified a maximum heat radiation intensity of 22 kW/m[2] to which LPG
thanks should be exposed (API, 1996). Performance and safety of LPG tanks exposed to radiant heat of
wildfires have been investigated by Heymes et al. (2013).
Despite the fact that the risk of radiant heat seems easier to quantify (than the risk of airborne embers)
based on current techniques and available databases, it is missing in the available directives and guidelines.
For instance, the FireSmart®, a Canadian field guide for protecting oil and gas facilities against wildfires,
identifies a rule-of-thumb minimum safety distance of 3m for propane tanks (pressurized tank) from forest
vegetation (FireSmart, 2012). However, Heymes et al. (2013) showed that even a small fire of 2m high and
5m wide is able to increase the internal pressure of LPG tanks and eventually lead to a BLEVE and
subsequent fireball.
Wildfire-induced fires in the form of tank fires or pool fires can trigger secondary fires or explosions in
other process vessels and storage tanks, leading to a domino effect. Figure 10 shows fire propagation in a
fuel storage plant in Puerto Rico in 2009 which initiated from overspill and ignition of a gasoline storage
tank and propagated to other 21 storage tanks out of 40 (CSB, 2015).
To quantify the impact of a wildfire on an oil and gas facilities, the damage probabilities of the process
vessels exposed to the wildfire's radiant heat (i.e., the primary vessels) as well as the damage probability of
neighboring vessels exposed to the heat radiation of fires at the primary vessels need to be assessed. In this
regard, dose-response relationships which associate the damage probability of process vessels to the
intensity of received heat radiation can be used.

---

[5] Liquefied Petroleum Gas (LPG), mostly consisting of propane and butane, is a flammable substance used as fuel in heating, cooking, and vehicles.
[6] Boiling Liquid Expanding Vapor Explosion




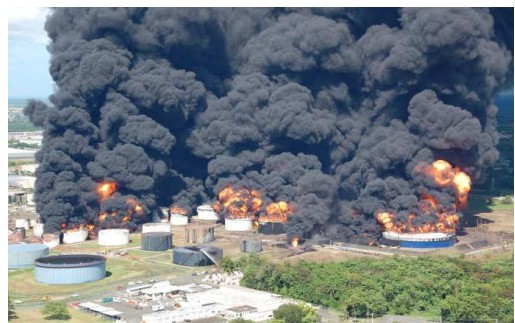


**Figure 10**. Fire domino effect in a tank farm in Puerto Rico in 2009 (CSB, 2015).
For instance, Cozzani et al. (2005) developed simplified probit functions to correlate the time to failure (ttf)
of vessels to their size and the intensity of received heat (a minimum required value of 15 kW/m² for
atmospheric vessels and 50 kW/m² for pressurized vessels). Equations (9)-(11) can be used to assess the
damage probability of atmospheric process vessels, including the storage tanks:
$\ln(ttf) = -1.13 \ln(Q_x) - 2.67 \times 10^{-5} V + 9.9$          (9)
$Y = 12.54 - 1.85 \ln(ttf)$          (10)
$P = \phi(Y - 5)$          (11)
where ttf (s) is the time to failure of the exposed vessel (due to wildfire's heat or a primary tank fire's heat);
$Q_X$ (kW/m²) is the received heat radiation by the vessel, calculated using Equation (9); V (m³) is the volume
of the vessel; Y is the probit value; P is the damage probability of the vessel; $\phi(.)$ is the cumulative standard
normal distribution. For the sake of exemplification, consider the hypothetical tank farm in Figure 11,
where atmospheric storage tanks T1 and T2 are exposed to the wildfire's radiant heat of greater than 15
kW/m² and may catch fire. Tank T3 is too far to damage directly by the wildfire's heat radiation but may
damage via a domino effect given wildfire-induced fires at T1 or T2.
Given the characteristics of the wildfire and the location of the tank farm (e.g., using Figure 4(b)) and the
distance of the storage tanks from the head fire, the amount of radiant heat received by T1 and T2 can be
calculated using Equations (7) and (8); accordingly, the conditional damage probabilities of the tanks given
the wildfire, i.e., P(T1|w) and P(T2|w), can be estimated using the probit functions given in Equations (9)-
(11). Given that the wildfire would ignite tank fires at either T1 or T2, three mutually exclusive domino
effect scenarios can be envisaged in which T3 would damage and catch fire from either T1 or T2 (Figure

429      12).

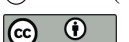



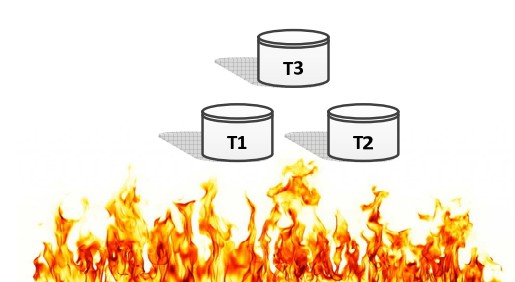


**Figure 11**. A hypothetical case of three atmospheric storage tanks exposed to wildfire.


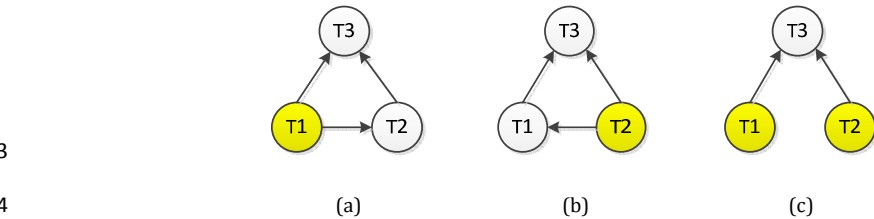

434         (a)         (b)         (c)

**Figure 12**. Wildfire-induced domino effect scenarios. Tanks directly impacted by the wildfire have been denoted by color yellow.

As a result, $P(T3|w)$ can roughly be estimated as the aggregation of the three domino effect scenarios as
$P(T3|w) = P(T3|w)_a + P(T3|w)_b + P(T3|w)_c$, where:
• Figure 12(a): $P(T3|w)_a = P(T1|w) . (1 - P(T2|w)) . \{P(T3| T1) \cup \{P(T2| T1) . P(T3| T2)\}\}$
• Figure 12(b): $P(T3|w)_b = (1 - P(T1|w)) . P(T2|w) . \{\{P(T1| T2) . P(T3| T1)\} \cup P(T3| T2)\}$
• Figure 12(c): $P(T3|w)_c = P(T1|w) . P(T2|w) . \{P(T3| T1) \cup P(T3| T2)\}$.
Similar to $P(T1|w)$ and $P(T2|w)$, the conditional probabilities $P(T1| T2)$, $P(T2| T1)$, $P(T3| T1)$, and $P(T3| T2)$ can be estimated using probit functions in Equations (9)-(11) based on the amount of heat radiation a secondary tank receives from fire at a primary tank. Having the conditional damage probabilities of the storage tanks (conditioned on the occurrence of a wildfire of given characteristics), the marginal damage probabilities, e.g., for T3, can be calculated as $P(T3) = P_w . P(T3|w) = P_I . P_B. P(T3|w)$.
For large oil and gas facilities with many process vessels of different type and dimension, in which complicated interaction among the process vessels would not allow a manual calculation of damage




probabilities, more sophisticated techniques such as Bayesian network (Khakzad et al., 2013) can be
employed.

## 5. Conclusions

The present study has been inspired by recent massive wildfires in the Province of Alberta, Canada,
jeopardizing the operation and safety of oilsands facilities as a key contributing factor to the nation's
economy. Despite the extensive oilsands operations in Canadian wildlands and an ever-increasing risk of
wildfires, mainly due to global warming, quantitative methodologies for assessing and managing the risk of
wildfires in the context of natural-technological accidents (i.e., technological accidents triggered by natural
disasters) are lacking.
In the present study, we made an attempt to develop a dynamic risk assessment methodology for wildfire-
prone process plants by integrating the Canadian online wildfire information system and available QRA
techniques. Since the wildfire information system is updated on a daily basis providing forecasts for the
same day and the next day, the developed methodology can help facilities owners and safety managers
predict the risk of wildfires at least a day ahead of time and thus devise appropriate protection and
mitigation measures.
In most of wildland oil and gas facilities, the separation distances (buffer zones) between oil facilities and
forest vegetation are usually determined based on approximate analyses (e.g., in Canada it is based on
FireSmart® guidelines). As such, the developed methodology can be employed not only in risk-based
identification of more dependable buffer zones but also in design of the layout of oil facilities so as to
increase their robustness against wildfire-induced domino effect scenarios.

## Appendix

Identification of view factor in solid flame model

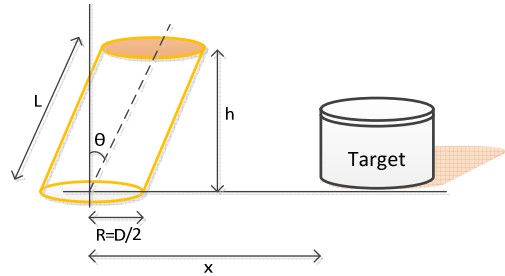




**Figure 13**. Flame as a tilted cylinder

$F_{view}$ can be calculated as a function of vertical $F_v$ and horizontal $F_h$ view factors as (Assael and Kakosimos,

477     2010):

$F_{view} = \sqrt{F_v^2 + F_h^2}$
where:
$\pi F_v = -E \tan^{-1}\emptyset + E\left[\frac{\alpha^2+(\beta+1)^2-2\beta(1+\alpha \sin\theta)}{AB}\right]\tan^{-1}\left(\frac{A\emptyset}{B}\right) + \frac{\cos\theta}{C}\left[\tan^{-1}\left(\frac{\alpha\beta-F^2\sin\theta}{FC}\right) + \tan^{-1}\left(\frac{F \sin\theta}{C}\right)\right]$
$\pi F_h = \tan^{-1}\left(\frac{1}{\emptyset}\right) + \frac{\sin\theta}{C}\left[\tan^{-1}\left(\frac{\alpha\beta-F^2 \sin\theta}{FC}\right) + \tan^{-1}\left(\frac{F \sin\theta}{C}\right)\right] - \left[\frac{\alpha^2+(\beta+1)^2-2(\beta+1+\alpha\beta \sin\theta)}{AB}\right]\tan^{-1}\left(\frac{A\emptyset}{B}\right)$
$\alpha = \frac{L}{R}$
$\beta = \frac{X}{R}$
$A = \sqrt{\alpha^2 + (\beta + 1)^2 - 2\alpha(\beta + 1)\sin\theta}$
$B = \sqrt{\alpha^2 + (\beta - 1)^2 - 2\alpha(\beta - 1)\sin\theta}$
$C = \sqrt{1 + (\beta^2 - 1)\cos^2\theta}$
$\emptyset = \sqrt{(\beta - 1)/(\beta + 1)}$
$E = \frac{\alpha \cos\theta}{\beta - \alpha \sin\theta}$
$F = \sqrt{\beta^2 - 1}$
The angle of tilt, θ, can be calculated as a function of wind speed $u_w$ as (Pritchard and Binding, 1992):
$\frac{\tan\theta}{\cos\theta} = 0.666 \, Fr^{0.333} Re^{0.117}$
where Fr is the Froud number $Fr = \frac{u_w^2}{g\emptyset}$, and Re is the Reynolds number $Re = \frac{u_w \rho_a \emptyset}{\eta_a}$, both non-dimensional
numbers. $\rho_a$ and $\eta_a$ are, respectively, the density ($\sim$ 1.21 kg/m³) and viscosity ($\sim$ 16.7 µ Pa s) of air; g is
gravitational acceleration ($\sim$ 9.81 m/s²).





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
