# Peer review of "Assessing the impact of wildfire on Canada's oilsands facilities"

_Natural Hazards and Earth System Sciences, 2018_

## Referee Comment (RC1) · Anonymous Referee #1 · 21 Aug 2018

General comments The paper mainly presents a literature review about risk assessment for wildfire with a particular concern for the risk of fire of oil and gas facilities. The author is particularly interested by the region of Alberta Canada. To present the context, the author reproduces the maps of the main variables of interest (fire weather index, fire danger index, fire rate of spread, head fire intensity) for a particular fire occurred on May, 2016 for Canada as a whole, as estimated by the Canadian fire authority. The author also presents the ignition probability map elaborated by the Canadian fire authority, for Alberta (not for Canada). Then the general model of risk estimation is presented, based on the wildfire probability estimation, using the ignition probability and the burn probability. Some empirical and semi empirical models are presented for burn probability, as examples. Then the author presents head fire intensity (HFI) models, in a review form. In fact the author will finally focus on the estimation of the area-fire or reaction intensity, which depends on HFI. The reaction intensity at a distance x from

the flame center is the main variable of interest to study the risk for oil facilities because it is the explanatory variable of both the time to failure of the exposed vessel and the damage probability of the vessel. The model of the conditional damage probability is presented for a simplified and hypothetical case consisting of three tanks (domino effect scenarios). While the paper in generally well written and well documented, I think that the literature review is too long in comparison to the scope of the paper and to the final objective which is the calculation of the conditional damage of the vessels. The reader who is not aware about wildfire risk models may learn a lot about these models by reading the document. However, a reader who knows these models will have a deception especially that the case study is simplistic and with no application. My detailed comments and questions are listed in the following:

- Lines 65-67 this sentence should outline the methodology adopted in the cited paper. It is important to mention that high-hazard resolution and demographic projections were used under a business-as-usual scenario of greenhouse gas emissions. The referred paper diagnostics is that by 2100, 2/3 of the European population will be affected annually by 2100. The author should correct the sentence to be close to what is said in the reference https://www.thelancet.com/journals/lanplh/article/PIIS2542-5196(17)30082-7/fulltext . A comment was addressed to this paper https://www.thelancet.com/pdfs/journals/lanplh/PIIS2542-5196(17)30084-0.pdf outlying that people should be aware about the assumptions of the study (constant human vulnerability for the projection period, equal to present vulnerability). So the author should also take this in consideration. The method of estimating human costs is very important to notice. Finally I don't know if "affect" (the word used by the author) is the same as 'to be exposed" which is the terminology of Forzieri et al. (2017). - Lines 77-78. I would say that the sentence is not related to the content of the referenced papers; the papers that are referenced are not dealing with the same scopes and methods; some are operational and others methodological. Some are about risk assessment and others about risk management. Some are about projections within climate change others are about assessment of observed fires

in the field. Thus, they need to be considered separately; the sentence should be reformulated with emphasizing on the specificity of each referenced paper. - Line 79 The author should mention the document "fireSmart for the oil and gas industry" http://www.enform.ca/files/Comms_Uploads/events/events_psc/2017/Speaker_presentations/PSC_Wildfire_2017_May3_0 - Line 127. What do author means by "in the absence of methodology to quantify the risk"? The sentence should be reformulated. - Line 130 In Heymes et al. (2013) study the distance is 50 m. Is what the author means when using "such"? - Line 146 "... fuel (trees...shrubs)'. This parenthesis should be presented earlier in the text, when using fuel for the first time. - Line 155. The reference is without date. - Line 156 In Figure 3a, the scale needs to be given. - Lines 156-160 In the text, it must be clear (like in Line 176) that this was obtained by the source indicated under Figure 3a and 3b. - Lines 167-173 the key word PROMETHEUS should be included in the text. Also it should be noticed that the model is based on Huygens principle of wave propagation. - Lines 186-187 the author should briefly specify what are the main common points and differences between these models (Latham, Lawson, Anderson). Is the title of the paragraph complete or is it necessary to add "Canadian"? - Line 192 A new paragraph should start from "Using the developed.." - Line 194. Author need to justify considering severity in monetary units. Why "preferably" knowing that it is a lot of work and assumptions to convert direct damages in monetary units? - Line 195 Author should use the word severity - Line 198 Author needs to justify the choice of 400 m$^2$. Is it in linkage with the spatial scale evocated in the burning probability? See for example the glossary https://www.nrfirescience.org/sites/default/files/documents/ScottGlossaryWildlandFireTerms.pdf - Line 219 It should be noticed that satellite data were used in Jurdao et al. paper. - Line 221 Drought code is not a parameter. Author needs to reformulate. - Lines 256-258. Do the cited Papadopoulos et Pavlidou (2010) review includes sophisticated models only? The sentences should be reformulated. - Line 260 Author should justify why to detail about the model of Drossel et Schwabl which is older than others previously referenced. - Line 263 Moore's neighborhood. It should be referenced (it is well explained in https://bib.irb.hr/datoteka/278897.Ljiljana_Bodrozic_ceepus2006_2.pdf)

- Line 271 author should give a reference for combining with Monte Carlo simulations. It is also needed to justify why using Monte Carlo simulations. - Line 284 North should be indicated in Figure 6. - Line 287 Do Monte Carlo simulations continue to be adopted in the literature? - Line 288 –density of trees). Do author means only density of trees? - Line 305 Un more recent paper by Cheney might be referenced for grass land case such as Cheney et al. (1998) https://www.researchgate.net/publication/248885594_Prediction_of_Fire_Spread_in_Grasslands - Line 320 Is there a difference between guidelines for British Colombia and Alberta? It is not said. - Line 325 On which model is based the map of Figure 4b?. Is it different from the models presented in Eq. (5) and (6)? It is not clear. What is the scale (mesh) for this map (Figure 4b)? Is downscaling needed for operational purposes? if 'Figure 4b can be used directly" and what about Figure 9? - Line 417. Maybe there is a mistake in the text because $Q_x$ is an explanatory variable in Eq. (9). Maybe author means Eq. (8)? - Line 415 what is the normal variable linked to p has to be linked to ïĄę? - Line 450 the author should explain the novelty of the actual submission with respect to his previous paper (Khazad et al. 2018) https://www.researchgate.net/publication/325995779_Quantitative_assessment_of_wildfire_risk_in_oil_facilities - Line 459 It is not clear from the text where the presented methodology is dynamic. In this paper author did not use up-dating of information. The sentence should be reformulated especially within the content in Line 462 (can help). - Line 467. Do the author means that it is a perspective to use the model to optimize or rationalize the buffer zones? The word perspective is missing.

Please also note the supplement to this comment:
https://www.nat-hazards-earth-syst-sci-discuss.net/nhess-2018-149/nhess-2018-149-RC1-supplement.pdf
* * *

---

## Referee Comment (RC2) · Anonymous Referee #2 · 23 Oct 2018

The title is of today's interest for both academy and industry. The manuscript is well organised and it can be considered for the potential publication in the journal in its current condition.

---

## Author Comment (AC1) · 25 Oct 2018

Thank you very much for your time and positive feedback.

---

## Author Comment (AC2) · 25 Oct 2018

Thank you for your time and detailed comments. Here are the response to the comments; the manuscript will be revised accordingly.

General comments

C1: The author also presents the ignition probability map elaborated by the Canadian fire authority, for Alberta (not for Canada).

Re: The study's focus is on Canadian oilsands operations which are concentrated in Alberta. Among the three main oilsands fields in Alberta, i.e., Athabasca, Peace river, and Cold Lake, only a very small part of Cold Lake's extends in Saskatchewan. That is why the ignition probability map of Alberta has been used.

C2: I think that the literature review is too long [. . .]. The reader who is not aware about

wildfire risk models may learn a lot about these models by reading the document [. . .] especially that the case study is simplistic and with no application.

Re: Compared to other types of Na-techs, the ones triggered by wildfires have received less attention from process safety researchers and practitioners. This to some extent justifies the explanatory introduction (literature review) although we don't really think a 3-page introduction, one page out of which is devoted to a table and a figure, is too long. The case study has purposely been chosen simple to help the readership focus on the wildfire as the issue of wildfire-induced domino effects in the chemical and process plants has already been studied in the previous studies (e.g., https://www.researchgate.net/publication/325995779_Quantitative_assessment_of_wildfire_).

Detailed comments

C3: Lines 65-67 this sentence should outline the methodology adopted in the cited paper [. . .] Finally I don't know if "affect" (the word used by the author) is the same as 'to be exposed" which is the terminology of Forzieri et al. (2017).

Re: Lines 65-67 refer to the online newsletter published by European Joint Research Centre (the link below) where it says "Weather-related disasters could affect around two-thirds of the European population annually by the end of this century" with a reference to Forzieri et al. (2017). https://ec.europa.eu/jrc/en/news/europe-be-hit-hard-climate-related-disasters-future In order to avoid any misunderstanding, this part is now changed to "Rising temperatures and climate change have increased the risk of weather-related hazards in Europe (European Joint Research Centre, August 2017)."

C4: Lines 77-78. I would say that the sentence is not related to the content of the referenced papers [. . .] some are operational and others methodological. Some are about risk assessment and others about risk management.

Re: Thanks; we just kept Preisler et al. (2004) and Scott et al. (2012; 2013) as examples of wildfire risk assessment studies.

Interactive
comment

C5: Line 79 The author should mention the document "fireSmart for the oil and gas industry. Re: Thanks; FireSmart is now added.

C6: Line 127. What do author means by "in the absence of methodology to quantify the risk"? The sentence should be reformulated.

Re: Thanks; it is now changed to "In the absence of quantitative methodologies for risk assessment of wildfires in wildland-industrial interfaces".

C7: Line 130 In Heymes et al. (2013) study the distance is 50 m. Is what the author means when using "such"?

Re: Not necessarily. Heymes et al. (2013) did the experiment for a pressurized tank 50m away from the wildfire to find that the tank would receive a heat radiation of 26 kW/m2 which is below the damage threshold (the conclusions section of their paper). (i) Knowing that atmospheric tanks are more vulnerable than pressurized tanks (damage threshold = 15 kw/m2), then a 50 m safety distance would not be sufficient. (ii) As mentioned on Pg. 101 of their paper, this distance is 25 m in Spain and 30 m in the U.S.

C8: Line 146 "... fuel (trees...shrubs)'. This parenthesis should be presented earlier in the text, when using fuel for the first time.

Re: Line 146 is the first time the word "fuel" is used in the manuscript.

C9: Line 155. The reference is without date.

Re: The reference is a website. But in the reference list we added the date of last checking.

C10: Line 156 In Figure 3a, the scale needs to be given.

Re: The scale is given in the figure's legend (0-30), but it is now added to the text.

C11: Lines 156-160 In the text, it must be clear (like in Line 176) that this was obtained

by the source indicated under Figure 3a and 3b.

Re: Your comment is not clear; we have already mentioned Figures 3(a) and (b) in lines 156-160.

C12: Lines 167-173 the key word PROMETHEUS should be included in the text. Also it should be noticed that the model is based on Huygens principle of wave propagation.

Re: Thanks; it is now added with some brief explanation: "Prometheus is a deterministic wildland fire growth simulation model based on Huygens principle of wave propagation. It uses the Fire Weather Index (FWI) and Fire Behaviour Prediction (FBP) sub-systems of the Canadian Forest Fire Danger Rating System."

C13: Lines 186-187 the author should briefly specify what are the main common points and differences between these models (Latham, Lawson, Anderson). Is the title of the paragraph complete or is it necessary to add "Canadian"?

Re: Thanks; it was made clear that the work of Latham and Schlieter (1989) and Anderson (2002) are about lightning-induced ignitions whereas the work of Lawson (1994) about human-induced ignitions. "Canadian" was added to the tile.

C14: Line 192 A new paragraph should start from "Using the developed."

Re: Thanks; we did so.

C15: Line 194. Author need to justify considering severity in monetary units. Why "preferably" knowing that it is a lot of work and assumptions to convert direct damages in monetary units?

Re: "Monetary units" because the aim is to perform a quantitative risk assessment. With non-monetary units we would have no choice but to carry out a qualitative or semi-qualitative risk assessment, for instance, using a risk matrix, which would be too subjective.

C16: Line 195 Author should use the word severity.

Re: Considering the response to the previous comment, the result would be the risk in monetary amounts (quantitative) not severity (qualitative).

C17: Line 198 Author needs to justify the choice of 400 m2. Is it in linkage with the spatial scale evocated in the burning probability? See for example the glossary https://www.nrfirescience.org/sites/default/files/documents/ScottGlossaryWildlandFireTerms

Re: Thanks; 400 m2 has been chosen according to Preisler et al. (2004) which consider a small fire as a fire in an area less than 0.04 hectare. We added this to Line 220.

C18: Line 219 It should be noticed that satellite data were used in Jurdao et al. paper.

Re: Thanks; it is now added.

C19: Line 221 Drought code is not a parameter. Author needs to reformulate.

Re: Thanks; it is now corrected.

C20: Lines 256-258. Do the cited Papadopoulos et Pavlidou (2010) review includes sophisticated models only? The sentences should be reformulated.

Re: In Line 257, "Less sophisticated models" does not have to do with Papadopoulos et Pavlidou (2010); it's making a comparison between the forest-fire model of Drossel and Schwabl (1992) in Line 260 with the previous complex models as FARSITE and BehavePlus in Line 254.

C21: Line 260 Author should justify why to detail about the model of Drossel et Schwabl which is older than others previously referenced.

Re: There was no specific reason except the model is simple and does not need much input information. In order not to give the impression that it is more important than other recent models, its detailed explanation is removed.

C22: Line 263 Moore's neighborhood. It should be referenced (it is well explained in

https://bib.irb.hr/datoteka/278897.Ljiljana_Bodrozic_ceepus2006_2.pdf)

Re: We removed the forest-fire model's explanation; no further need to reference Moore's neighborhood.

C23: Line 271 author should give a reference for combining with Monte Carlo simulations. It is also needed to justify why using Monte Carlo simulations.

Re: We could not find an explicit application of Monte Carlo except our previous study, Khakzad et al. (2018). But we got the idea from the work of Finney (2002): "Examples include fire-risk assessment that require thousands of fire simulations from various ignition points and fuel treatment optimization". (Pg. 1423). Finney MA. (2002). Fire growth using minimum travel time methods. Can. J. For. Res. 32: 1420-1424. We added the reference to Line 271.

C24: Line 284 North should be indicated in Figure 6.

Re: Thanks; it is added.

C25: Line 287 Do Monte Carlo simulations continue to be adopted in the literature?

Re: Please see the response to Comment 23.

C26: Line 288 –density of trees). Do author means only density of trees?

Re: Not only the density, because, for instance, "The model assumes there is no wind. So, the fire must have trees along its path in order to advance. That is, the fire cannot skip over an unwooded area (patch), so such a patch blocks the fire's motion in that direction." (https://ccl.northwestern.edu/netlogo/models/Fire).

C27: Line 305 Un more recent paper by Cheney might be referenced for grass land case such as Cheney et al. (1998).

Re: Thanks, it is now added.

C28: Line 320 Is there a difference between guidelines for British Colombia and Alberta? It is not said.

Re: No, there is not. Please see the response to Comment 29.

C29: (i) Line 325 On which model is based the map of Figure 4b?. (ii) Is it different from the models presented in Eq. (5) and (6)? It is not clear. (iii) What is the scale (mesh) for this map (Figure 4b)? (iv) Is downscaling needed for operational purposes? (v) if 'Figure 4b can be used directly" and what about Figure 9?

Re: (i) & (v) Prometheus is used as the Canadian wildland fire growth simulation model by the Canadian Interagency Forest Fire Centre and its member states, that is, all the Canadian provinces and territories except Nunavut. So, Figures 4(b) and 9 are the results of the same model. Since Prometheus is a quantitative model, Fig. 4(b) seems to be the direct output of the model whereas Fig. 4(b) seems to have been explained in classes for a better communication with public (http://wildfire.alberta.ca/).

(ii) In tha absence of Fig. 4(b), when the fire intensity is identified qualitatively as the classes (e.g., Fig. 9) or the height of the flames could be estimated visually, then Equation (6) can be used to estimate to quantify the fire intensity.

(iii) According to Tymstra et al. (2010), a raster representation of the wildland with 25m cells are used in Prometheus with ignition points as circles with a 0.5m diameter.

(iv) Further, three interpolation techniques have been provided in the model to produce optional raster fire behavior outputs.

C30: Line 417. Maybe there is a mistake in the text because Qx is an explanatory variable in Eq. (9). Maybe author means Eq. (8)?

Re: Thanks; it is now corrected to Eq. (8).

C31: Line 415 what is the normal variable linked to p has to be linked to ïAËŻeËŻ?

Re: The comment is not clear, but we checked the equation, it is correct.

C32: Line 450 the author should explain the novelty of the actual submission with respect to his previous paper (Khazad et al. 2018).

Re: Considering natural-technological accidents (Na-Techs) as a bow-tie diagram, the natural disaster and its characteristics belong to the left hand side while the impact of the natural disaster on the industrial plants and the subsequent accidents to the right hand side of the bow-tie.

In our previous work, we put more emphasis on the impact of wildfire on process plant, where a dynamic Bayesian network was used to estimate the probability of fire spread inside the plant. In other words, the focus in Khakzad et al. (2018) was more on the right hand side of the Na-tech bow-tie.

The focus of the present study, however, is on the wildfire by developing a methodology for quantification of its probability and intensity in wildland-industrial interfaces (the left hand side of the bow-tie).

C33: Line 459 It is not clear from the text where the presented methodology is dynamic. In this paper author did not use up-dating of information. The sentence should be reformulated especially within the content in Line 462 (can help).

Re: The proposed methodology is dynamic as the wildfire information is updated on a daily basis and so is the amount of calculated risk. However, since it is the input information that is updated not the risk assessment technique, we deleted the word "dynamic" from Line 459.

C34: Line 467. Do the author means that it is a perspective to use the model to optimize or rationalize the buffer zones? The word perspective is missing.

Re: The comment is not clear. However, the sentence was modified to "quantitative methodologies similar to the one presented in this study can be used to [. . .]".

[Figure]

2018-149, 2018.